# Inversion Method Characterization of Graphene-Based Coordination Absorbers Incorporating Periodically Patterned Metal Ring Metasurfaces

**DOI:** 10.3390/nano10061102

**Published:** 2020-06-02

**Authors:** Zhiyu Bao, Yang Tang, Zheng-Da Hu, Chengliang Zhang, Aliaksei Balmakou, Sergei Khakhomov, Igor Semchenko, Jicheng Wang

**Affiliations:** 1School of Science, Jiangnan University, Wuxi 214122, China; zhi_yu_bao@163.com (Z.B.); TANGYANGgemma@163.com (Y.T.); huyuanda1112@jiangnan.edu.cn (Z.-D.H.); jcwang@jiangnan.edu.cn (J.W.); 2Departments of Optics and General Physics, Francisk Skorina Gomel State University, Sovetskaya Str. 104, 246019 Gomel, Belarus; balmakou@yandex.ru (A.B.); khakh@gsu.by (S.K.); isemchenko@gsu.by (I.S.); 3National Laboratory of Solid State Microstructures, Nanjing University, Nanjing 210093, China

**Keywords:** graphene, metamaterials, parameter inversion, perfect absorption

## Abstract

In this paper, we propose a tunable coordinated multi-band absorber that combines graphene with metal–dielectric–metal structures for the realization of multiple toward perfect absorption. The parametric inversion method is used to extract the equivalent impedance and explain the phenomena of multiple-peak absorption. With the change of the Fermi level, equivalent impedances were extracted, and the peculiarities of the individual multiple absorption peaks to change were determined. By changing the structure parameters of gold rings, we obtain either multiple narrow-band absorption peaks or a broadband absorption peak, with the bandwidth of 0.8 μm where the absorptance is near 100%. Therefore, our results provide new insights into the development of tunable multi-band absorbers and broadband absorbers that can be applied to terahertz imaging in high-performance coordinate sensors and other promising optoelectronic devices.

## 1. Introduction

Metamaterials are artificially structured materials that can have negative permittivity or/and permeability, which can be achieved by periodic metal–dielectric arrays [1,2,3,4]. With the advent of metamaterials, new feasibilities have been arisen for devices related to stealth technologies [5,6], hyperlenses [7], and high-sensitivity sensors [8]. Metamaterial-based absorbers (MMA) have gained wide attention since the first near-perfect metamaterial absorber was designed by N. I. Landy et al. [9]. MMAs are generally constructed by metal–dielectric–metal (MDM) multilayers. By adjusting the geometric or material parameters, the metamaterial can be tuned for different operating frequencies, which are crucial for actual manufacturing and different from sensors [10], filters [11], solar photovoltaic devices [12], etc. Nonetheless, the design of some devices based on metamaterials [13,14,15] have a common disadvantage: that is, the devices’ operating frequencies cannot be adjusted according to different requirements. This has become a major obstacle hindering the further development of MMAs toward tunable MMAs. To challenge this drawback, we here report that the implementation of a single layer of graphene [16] into properly designed MMA structures can achieve tunable operating frequencies.

Graphene is a relatively new two-dimensional sheet composed of carbon atoms of atomic thickness with unique quantum, thermodynamic [17], and optical properties [18,19]. Due to its special optical properties, relating, first of all, to the high transmittance of surface plasmons (SP) [20,21,22], light propagation controlling at sub-wavelength distances in the structure of metamaterials is providing a powerful tool in nanophotonics. Graphene surface plasmon resonance (SPR) is supported in a wide frequency range including infrared and terahertz bands [23,24,25,26] such that it is perfect for designing tunable active metamaterials. Moreover, being adaptable to the Fermi level tuned by applying an external voltage or chemical doping [27], graphene can optimize the operating frequencies of the whole graphene-based metamaterial devices at their fixed geometrical parameters [28,29,30,31,32]. Therefore, the combination of graphene with an MDM structure to design a dynamically adjustable perfect absorber seems is promising.

While the proposed novel, multi-band, and tunable selective absorber utilizes graphene as its key component, its base structure uses gold as a ground plane that reflects electromagnetic waves. Between the ground plane and graphene, a substrate layer of SiO_2_ is situated, while the top layer represents an array of different sized gold rings (see – 1). The proposed structure is simulated using the commercial finite element method (FEM) solver COMSOL Multiphysics with the purpose of obtaining a number of closely located near-perfect absorption peaks. For reliability purposes, an additional numerical simulation utilizing the method of inverse parameter extraction is carried out to extract effective parameters and explain the phenomenon of multiple absorption peaks. By adjusting the Fermi level of graphene, at least three absorption peaks can be tunable between 8 μm (37.5 THz) and 14 μm (21.4 THz) while the structural changes in parameters of metal rings lead to a single wide absorption peak with an absorptance higher than 80%. The position of the single peak is also Fermi-level adjustable.

## 2. Design of the Absorber and Its Theoretical Analysis 

The three-dimensional structure of the graphene-based selective absorber is proposed in Figure 1a indicating TM (Transverse Magnetic) waves in the incident *XZ* plane by red arrows. The yellow layer and rings represent gold with thicknesses of *d*_1_ and *d*_3_, respectively. The blue layer represents SiO_2_ with a thickness of *d*_2_. A single layer of graphene between the upper gold rings and the SiO_2_ substrate is pictured as a honeycomb structure. Within the operating wavelengths, the dispersion of dielectric constant of SiO_2_ [33] is negligible and therefore fixed at *ε*_SiO2_ = 3.9, and the permittivity of gold can be expressed as *ε* = 1 − ($\omega_{2_{p}}$/(*ω*^2^ + *i*ωγ)) using the Drude model [34] with the value of constant plasma frequency *ω_p_* = 1.36 × 10^16^ rad/s and collision frequency *γ* = 3.33 × 10^13^ rad/s. The graphene is simulated in COMSOL by setting the surface current density.

The Kubo formula describes the surface conductivity of graphene layers [35,36,37]:(1)$$\begin{array}{ll} \sigma_{gra} & =\sigma_{\mathrm{int}er}+\sigma_{\mathrm{int}ra}=\frac{2e^{2}k_{B}T}{\pi \hbar^{2}}\frac{i}{\omega +i/\tau }\ln\left[2\cosh(\frac{E_{f}}{2k_{B}T})\right]\\ & \;+\frac{e^{2}}{4\hbar^{2}}\left[\frac{1}{2}+\frac{1}{\pi }\arctan(\frac{\hbar \omega -2E_{f}}{2k_{B}T})-\frac{i}{2\pi }\ln\frac{{\left(\hbar \omega +2E_{f}\right)}^{2}}{{\left(\hbar \omega -2E_{f}\right)}^{2}+4{\left(k_{B}T\right)}^{2}}\right] \end{array}$$
where *σ*_inter_ and *σ*_intra_ are expressed as the interband and intraband transition contributions, respectively. *k_B_*, ℏ, and *e* represent the Boltzmann constant, reduced Plank constant, and electron charge, respectively. *ω* is the angular frequency of the incident radiation, *E_f_* is the graphene Fermi energy level, *τ* is the electron-phonon relaxation time, and *T* is the ambient temperature. As we only consider highly doped graphene, we should take into account that *E_f_* ≫ *k_B_T* and *E_f_* ≫ ℏ*ω*. Therefore, the Kubo equation can be simplified to a Drude-like equation [38,39,40]:(2)$$\sigma_{\mathrm{gra}}=\frac{e^{2}E_{f}}{\pi \hbar^{2}}\frac{i}{\left(\omega +i/\tau \right)},$$
where *τ* = *μE_f_*/(${ev}_{2_{F}}$), *E_f_* is set to 0.15 eV, the media carrier mobility *μ* is set to 1318 cm^2^ V^−1^ s^−1^ [41], and the graphene Fermi velocity *v_F_* is set to 1 × 10^6^ ms^−1^. As shown in Figure 1a, the periodic boundary conditions for the unit cell are along the *X* and *Y* axes. Figure 1b is the top view of the unit cell. The outer ring radii of the three rings are *R*, *R*_1_, and *R*_2_ respectively, and the widths are all *w*. The optimal parameters of the absorber that can be considered as default parameters are the following: *P*_x_ = 6.2 μm, *d*_1_ = 0.2 μm, *d*_2_ = 0.3 μm, *d*_3_ = 0.1 μm, *R* = 1.06 μm, *R*_1_ = 1.13 μm, *R*_2_ = 1.28 μm, and *w* = 0.44 μm.

The black curve in Figure 2a shows the absorptance spectrum for the absorber with default parameters. The obtained spectrum has three peaks of near-perfect absorption located at 9.24 μm, 10.3 μm, and 11.75 μm, respectively. It is important to investigate the influence of every individual group of rings *R*, *R*_1_, and *R*_2_ on the three absorptance peaks. Excluding all the rings from the structure except for (case A) – rings *R*, resulting in the green dotted line in Figure 2a with its peak at 9.54 μm; (case B) – rings *R*_1_, resulting in the red dotted line in Figure 2a with its peak at 10.21 μm; and (case C) – rings *R*_2_, resulting in the blue dotted line in Figure 2a with its peak at 11.61 μm. For the complex structure with a full set of rings in Figure 1, the absorptance is shown by the black curve in Figure 2a where all three separate peaks mentioned above take place. The purpose is to focus on the optimal parameters of the rings in a way that they can be excited individually (see Figure 2b–e) by the external radiation of three different frequencies, but at the same time, the sizes of rings are close to each other. Therefore, the peaks of absorptance are short-distanced and can be merged by applying specific external physical factors. Electric field distribution inside the rings in Figure 2b–e provides additional information facilitating precisely choosing the width of the rings, their outer coating, and the proper doping agent.

An inversion algorithm is applied to extract the normalized impedance of the multiple-peak absorber for explaining the physical mechanism of the phenomenon. The absorber is considered as a uniform layer with material parameters *μ*_1_ and *ε*_1_, which is surrounded by an outer layer with material parameters *μ*_0_ and *ε*_0_, as illustrated in Figure 3b. An incident wave, as before, propagates along the *Z* axis—that is, perpendicular to the surface of the absorber. For simplicity, layers 1 and 3 in Figure 3b represent air. The electromagnetic field strength in any conductive medium can be expressed as [42]:(3)$$\begin{array}{l} E_{x}=E_{x0}\cdot e^{-jkz}\\ E_{y}=\frac{k}{\mu \omega }\cdot E_{x0}\cdot e^{-jkz} \end{array}$$

When TM propagates in medium 1, without loss of generality, we set the amplitude *E*_x0_ = 1. Then, the electromagnetic field strength in medium 1 can be written as:(4)$$\begin{array}{l} E_{1x}=e^{-jk_{1}z}+\Gamma_{1}e^{jk_{1}z}\\ E_{1y}=\frac{k1}{\omega \mu_{0}}e^{-jk_{1}x}-\Gamma_{1}e^{jk_{1}z} \end{array}$$

Then, the TM wave moves through the interface of medium 1 and medium 2, and the electromagnetic field strength in medium 2 can be written as:(5)$$\begin{array}{l} E_{2x}=\tau_{1}e^{-jk_{2}z}+\tau_{1}\Gamma_{2}e^{jk_{2}z}\\ E_{2y}=\frac{k_{2}}{\omega \mu_{1}}(\tau_{1}e^{-jk_{2}x}-\tau_{1}\Gamma_{2}e^{jk_{2}z}) \end{array}$$

Finally, the TM wave enters the medium 3, and the electromagnetic field strength in medium 3 can be written as:(6)$$\begin{array}{l} E_{3x}=\tau_{1}\tau_{2}e^{-jk_{3}z}\\ E_{3y}=\frac{k_{3}}{\omega \mu_{0}}\tau_{1}\tau_{2}e^{-jk_{3}z} \end{array}$$
where *k*_1_ = *k*_3_ = *k*_0_ is the wave number in vacuum, *k*_2_ = *nk*_0_ is the wave number in the relative medium, *Γ*_1_ and *Γ*_2_ are the reflection coefficients between medium 1 and medium 2, and medium 2 and medium 3, respectively. *τ*_1_ and *τ*_2_ are the transmission coefficients between medium 1 and medium 2, and medium 2 and medium 3, respectively.

We let the interface of medium 1 and medium 2 be the origin—that is, *Z* = 0. According to the boundary conditions encountered by the electromagnetic wave at the discontinuous interface when *Z* = 0, we can obtain:(7)$$\left\{ \begin{array}{l} E_{1x}=E_{2x}\to 1+\Gamma_{1}=\tau_{1}+\tau_{1}\Gamma_{2}\\ E_{1y}=E_{2y}\to k_{1}(1-\Gamma_{1})=\frac{k_{2}}{\mu_{r}}(\tau_{1}-\tau_{1}\Gamma_{2}) \end{array}\right.,$$
and at *Z* = *d*, that is, at the interface between medium 2 and medium 3:(8)$$\left\{ \begin{array}{l} E_{2x}=E_{3x}\to \tau_{1}e^{-jk_{2}d}+\tau_{1}\Gamma_{2}e^{jk_{2}d}=\tau_{1}\tau_{2}e^{-jk_{3}d}\\ E_{2y}=E_{3y}\to \frac{k_{2}}{\mu_{r}}(\tau_{1}e^{-jk_{2}d}-\tau_{1}\Gamma_{2}e^{jk_{2}d})=k_{3}\tau_{1}\tau_{2}e^{-jk_{3}d} \end{array}\right.,$$

According to the above-mentioned boundary conditions that the electromagnetic wave meets on the discontinuous interface, the scattering parameters can be obtained:(9)$$\left\{ \begin{array}{l} S_{11}=\frac{\Gamma_{1}(1-{\left(e^{-jk_{2}d}\right)}^{2})}{1-{\left(\Gamma_{1}e^{-jk_{2}d}\right)}^{2}}\\ S_{21}=\frac{1-\Gamma_{1}{}^{2}}{1-{\left(\Gamma_{1}e^{-jk_{2}d}\right)}^{2}}e^{-jk_{2}d} \end{array}\right.,$$
where *Γ*_1_ = (Z − 1)/(Z + 1). Since we use gold as the reflective layer of the entire absorber, there is no transmission spectrum, so *S*_21_ = 0. According to the selection of the sign [43,44], we can get:(10)$$Z=\pm \sqrt{\frac{(1+S_{11}{)}^{2}-S_{21}{}^{2}}{(1-S_{11}{)}^{2}-S_{21}{}^{2}}}=\frac{1+S_{11}}{1-S_{11}},$$

In order to minimize reflection, the normalized impedance of the absorber should be close enough to the impedance of free space—that is, the closer the real part of the normalized impedance is to unity and the closer the imaginary part is to vanishing, giving rise to the greater the absorptance of the structure. As is shown in Figure 4, three vertical dotted lines represent three resonance wavelength positions, and the positions of the first and second vertical dotted lines correspond to the first and second peaks of the real part of the impedance, respectively. The normalized impedances obtained at resonance wavelengths of 9.24 μm, 10.3 μm, and 11.75 μm are 0.9052 + 0.0378j, 0.8524 + 0.0760j, and 1.2099 − 0.1022j, respectively. The information is useful during optimizing the internal structure of the absorber as well as its composition with the purpose of minimizing the transmittance.

The relationship between the normalized impedance and the energy of Fermi level is investigated in Figure 5 to predict the absorptance properties when the graphed doping level is changing. As the Fermi level increases, the real and imaginary parts of the normalized impedance decrease slightly, and the resonance wavelength appears blue shifted. From this, we can see that as the Fermi level increases, the resonance wavelength of the absorber should appear blue shifted.

## 3. Absorptance Variational Analysis

In order to explore the dependence of peak displacement on structural parameters, we carry out a variation analysis. With *R* increasing independently from 1.04 μm to 1.08 μm, *R*_1_ increasing independently from 1.11 μm to 1.15 μm, and *R*_2_ increasing independently from 1.27 μm to 1.30 μm, the resonance wavelengths of the corresponding absorption peaks are red-shifted by 1.2 μm, 0.4 μm, and 0.4 μm respectively (see Figure 6a–c). The peak height is decreased by 0.4% during the first and second variation and by 1.6% for the third one. With increasing the width of the rings, all three resonance absorption peaks are blue shifted (see Figure 6d), while the peak height is increased slightly for the first and second, and it is decreased slightly for the third absorption peak. Thus, independent peak localization control is achievable by ring size adjustment.

The analysis of the change in the carrier mobility u in Figure 7 shows that when u increases, the first two absorptance peaks increase slightly, and the third absorption peak decreases significantly. This behavior is different from some previous patterned graphene multi-band absorbers [31,45]. We speculate that there is an interaction between the resonance absorption of the three metal rings and the single-layer graphene, thus forming such a changing behavior.

Since the structure proposed by us lays a layer of graphene, we can adjust the Fermi level of graphene by applying a voltage across the graphene, thereby changing the resonance frequency of the absorption peak. The Fermi level energy *E_f_* variation analysis in Figure 8 shows that with increasing *E_f_* from 0.1 to 0.6, all three absorption peaks are blue shifted by 0.4 μm, 0.5 μm, and 0.88 μm, respectively, while the peak heights are reduced slightly for the first two peaks and are increased slightly for the third one. The enhancement of the third peak is explained by the substantial reduction of the real part of the normalized impedance when increasing *E_f_* (see Figure 5). Known from reference [46], the graphene plasma dispersion relation is:(11)$$k_{spp}=\hbar \omega_{r}^{2}/\left(2\alpha_{0}E_{f}c\right)=\frac{2\pi^{2}\hbar c}{\alpha_{0}E_{f}\lambda_{r}^{2}},$$
where *α*_0_ = *e*^2^/(*ℏc*) is the structural constant of graphene, and *λ_r_* is the polarization resonance wavelength of graphene. From the above formula, the resonance frequency of the plasmon surface of the graphene can be obtained: *f_r_* ∝ (*α*_0_*E_f_*/(2π^2^*ℏcL*))^1/2^. That is, as the Fermi level increases, the resonant frequency also increases. 

## 4. Dynamically Adjustable Narrowband/Broadband Absorber

For the technical applications that require one broad absorption band instead of three narrow peaks, a modified version of the ring-type absorber is proposed with structural parameters *R* = 1.06 μm, *R*_1_ = 1.24 μm, *R*_2_ = 1.26 μm, and *w* = 0.44 μm. In this case, the absorptance greater than 80% occurs between wavelengths 10.51 and 11.31 μm (see Figure 9a)—that is, the full width at half maximum for the absorption spectrum can reach up to 1.17 μm. The positioning of the peaks can be controlled within 0.76 μm by Fermi level energy *E*_f_ varying between 0.4 and 0.9 eV, as shown in Figure 9b.

## 5. Conclusions

We proposed a novel, multi-band, tunable selective absorber with ground plane, a graphene layer, and a complex combination of gold rings patterned on the graphene layers. Through continuous optimization of the structural parameters of the three metal rings, three perfect closely located absorptance peaks between 8 and 12 μm are achieved. A method for extracting the inversion parameters of the absorber is used for obtaining the effective impedance of the structure and explaining the appearance of the three absorption peaks. An influence of Fermi levels onto the absorptance peaks resonance frequency is also investigated. The simultaneous realization of a single narrowband and broadband (with 0.8 μm bandwidth) absorption peaks with an absorptance greater than 80% is achieved. We assume that our findings can potentially be useful in those practical applications relating to adjustable optical filters, splitters, modulators.

## Figures and Tables

**Figure 1 nanomaterials-10-01102-f001:**
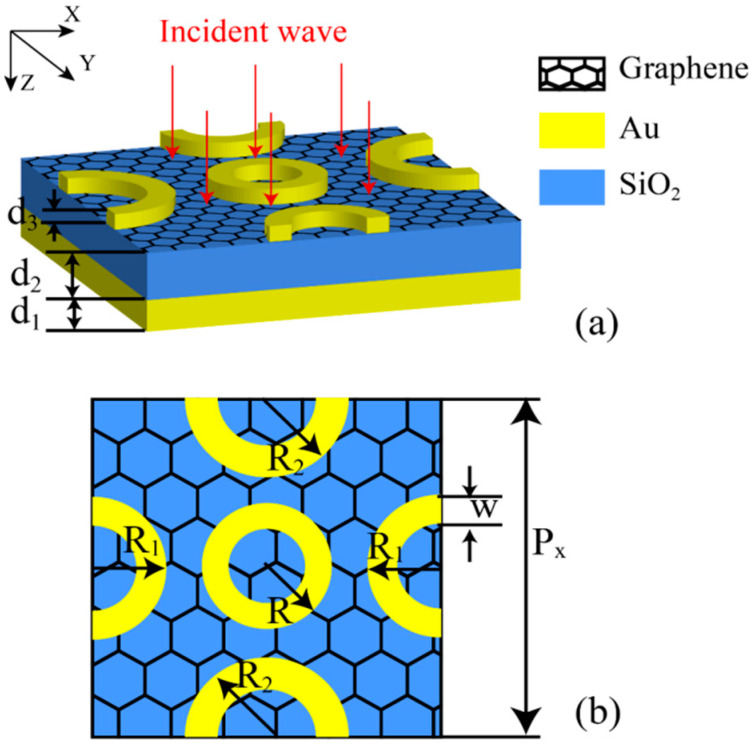
(**a**) The elementary unit cell of the proposed structure; (**b**) the top view of the unit cell. The width of the outer ring of the three metal rings is *R*, *R*_1_, and *R*_2_, and the width of the three rings is *w.* The thickness of the three rings is *d*_3_.

**Figure 2 nanomaterials-10-01102-f002:**
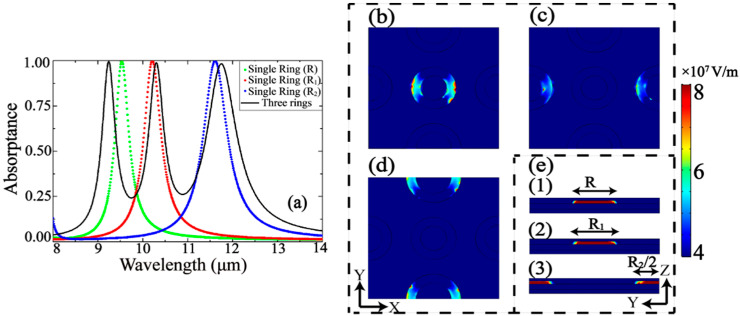
(**a**) Absorptance spectra for the absorber in Figure 1 with complete or partial ring structure; (**b**–**e**) represent the top and side views of the electric field distribution inside the absorber structure at wavelength of excitation 9.24 μm, 10.3 μm, and 11.75 μm, respectively.

**Figure 3 nanomaterials-10-01102-f003:**
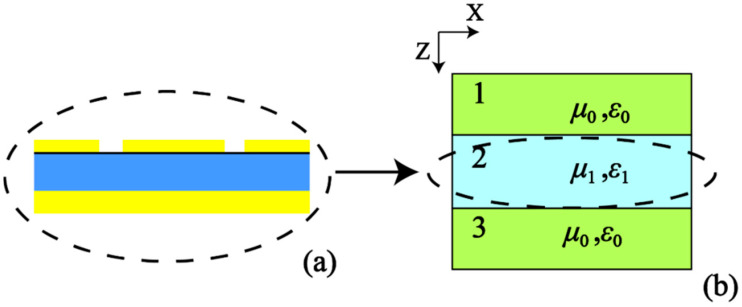
(**a**) Lateral schematic view of the absorber; (**b**) Equivalent model of the absorber.

**Figure 4 nanomaterials-10-01102-f004:**
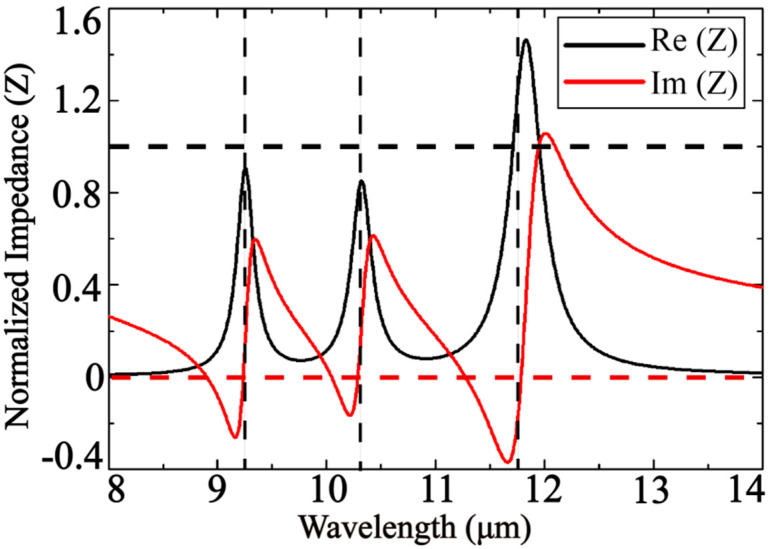
Normalized impedance obtained by inversion algorithm.

**Figure 5 nanomaterials-10-01102-f005:**
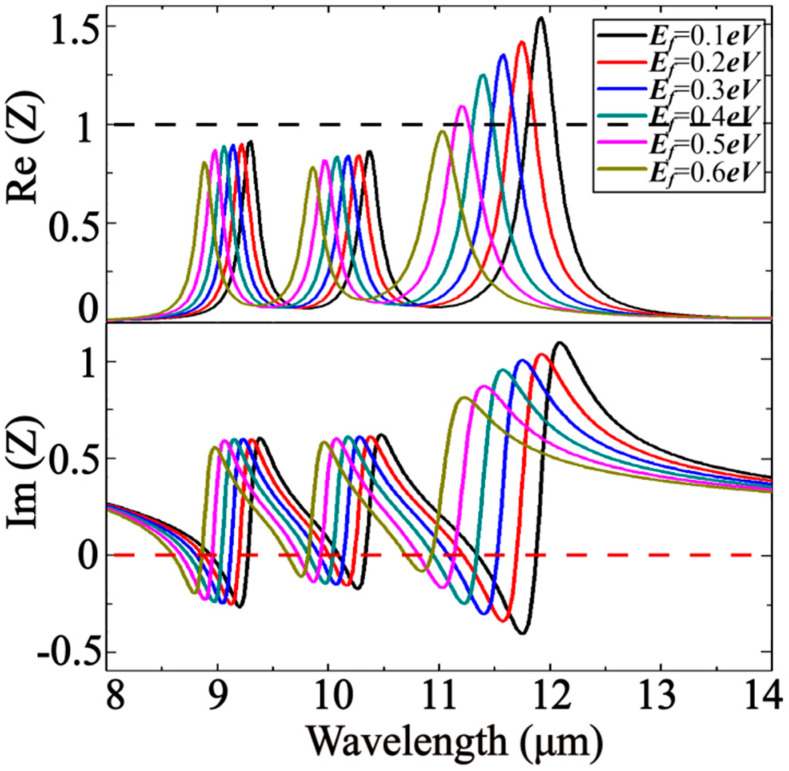
Normalized impedance as a function of Fermi level.

**Figure 6 nanomaterials-10-01102-f006:**
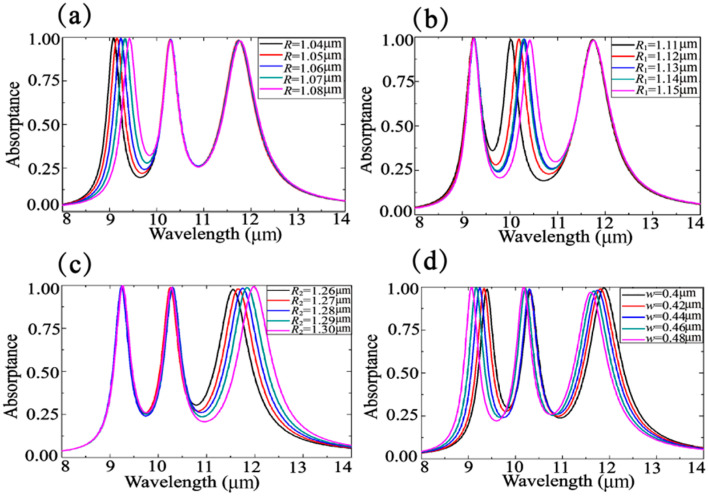
(**a**–**d**) Absorption spectra analysis with varying R, R_1_, R_2_, and *w*, respectively. Other parameters are set as default.

**Figure 7 nanomaterials-10-01102-f007:**
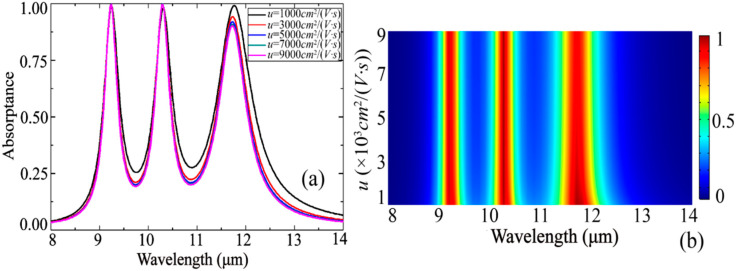
Carrier mobility variation analysis, (**a**) one-dimensional line diagram; (**b**) two-dimensional color diagram.

**Figure 8 nanomaterials-10-01102-f008:**
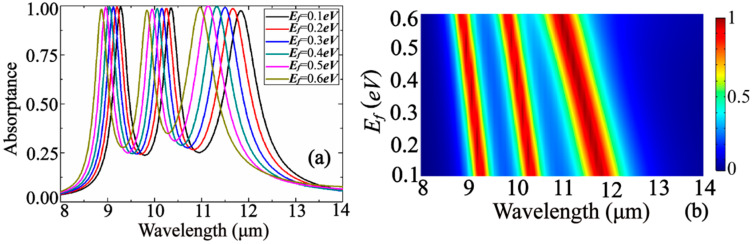
Fermi level energy variation analysis, (**a**) one-dimensional line diagram; (**b**) two-dimensional color diagram.

**Figure 9 nanomaterials-10-01102-f009:**
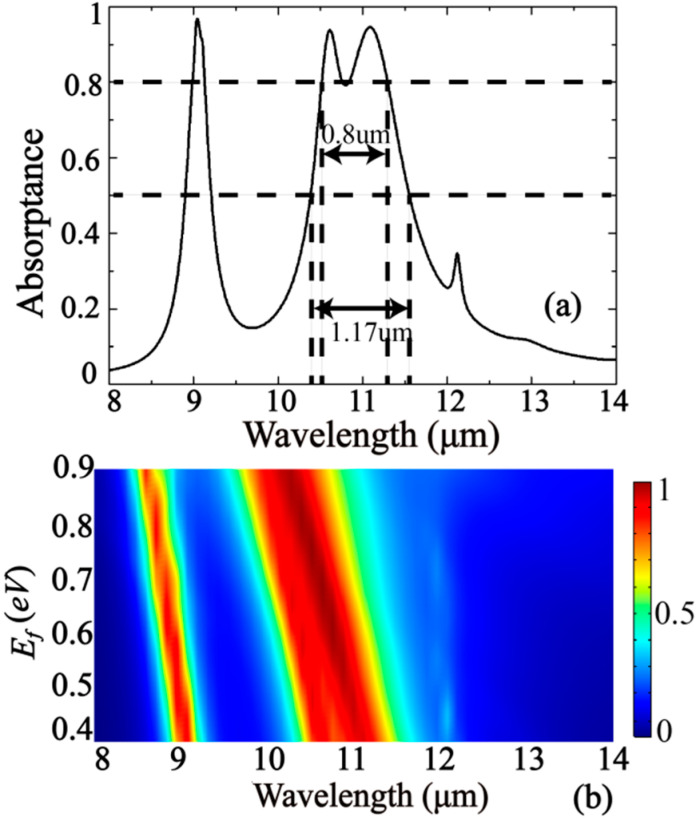
(**a**) The absorption spectrum for a modified absorber with the second and third peaks merged; (**b**) Fermi energy level variation analysis.

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
