# Peer review of "Inversion Method Characterization of Graphene-Based Coordination Absorbers Incorporating Periodically Patterned Metal Ring Metasurfaces"

_nanomaterials, 2020, doi:10.3390/nano10061102_

Round 1
Reviewer 1 Report
Article review
“Inversion method characterization of graphene-based coordination absorbers incorporating periodically patterned metal ring metasurfaces”
In this article, the authors have investigated a multiple perfect absorption via the multi-band absorber that combines graphene with metal-dielectric-metal structures. The authors have implemented a single layer of graphene into properly designed metasurface structures to tune operating frequencies.
In theoretical analysis, it says, the graphene was simulated by setting the surface current density. Theory-wise, it is difficult to consider a single graphene so that many numerical methods use an alternative way to consider the graphene. Here, the authors used the surface current density (Drude-like equation) for highly doped graphene. Then, the conductive medium theory was applied to obtain relevant parameters such as scattering parameters. The most important part in this article is the simulation absorption spectrum by resonance effect and surface-based plasmonic resonance.
- I wonder if all the equations suggested here are needed to present since too many irrelevant theoretical equations distract readers to focus the essence of theoretical background.
- The authors did not suggest any experimental data, which of course is predictable by numerical simulation but can significantly enhance the quality of the manuscript. The authors should at least rationalize their simulation work if they can’t add some convincing experimental data that can back up the theory.
- The important problem is the poor quality of English language used in this manuscript. This makes significant parts of the work unclear and difficult to understand. First of all, the language quality should be thoroughly enhanced.
- In the schematic drawing (Fig. 1), the inner and outer radii of the rings should be given clearly. Is R and R1 the same? The usage of”geometric parameter” is unclear tome, and this is a crucial in describing the designed device.
- It should be described how the independent tuning of the Fermi levels of the rings may be achieved in the suggested device.
Reviewer 2 Report
Review of the manuscript "Inversion method characterization of graphene-based coordination absorbers incorporating periodically patterned metal ring metasurfaces"
The manuscript consists of a theoretical description of a meta surface made from a gold ground plane, a dielectric layer, a monolayer of graphene and gold rings of three different sizes. The geometrical values are chosen this way, that three distinct absorption peaks appear in the simulation. If the geometrical values are chosen to be in closer agreement for all rings the absoption peaks start to merge into a broader absorption band. A tuning of the properties beside the fixed choice of the geometry is proposed by changing the Fermi of the graphene layer. The authors use a commercial FEM solver and a calculation of the complex impedance of the layer stack in parallel to verify the results. Basically this a good approach, since an analytical calculation of a simplified model can give valuable insights into the physical mechanisms. This explanation should be extended a bit in the manuscript. Additionally it is not always clear if the FEM model or the inversion method is used to study the influence of the different parameters.
Since a lot of papers these days deal with the calculation of meta-surfaces for THz applications, the authors should strengthen their discussion of the graphene layer and the tuning effect caused by it. A description of the main physical mechanisms behind this effect would upvalue the paper.
In the following I list additional remarks, comments, and questions regarding the manuscript:
(1) Page 1, Line 15: "perfect absorption", i.e., 100% seems to be a bit exaggerated, even when discussion absorption increases and decreases later in the manuscript. Maybe "towards perfect absorption" or something similar could be more appropriate.
(2) Page 1, Line 27: "with" should be "which can have", since metamaterials can also have positive values.
(3) Page 3, Line 84: What is "v_2_F" only "v_F" is given later?
(4) Page 3, Lines 88f: Where does these "optimal parameters" came from? Was there an optimization step? How was it done? What was the aim of the optimization (e.g. certain wavelengths)?
(5) Page 4, Lines 98ff.: The frequencies of the absorption peaks are given here, was the structure designed with this frequencies in mind, or are they just the result of the chosen geometry? Can an absorber be designed for certain frequencies without doing extensive simulations with a parameter sweep of the ring sizes?
(6) What was the motivation to choose a structure with three different geometries, that produce three absorption peaks? Can this approach extended to a higher number of peaks be adding more ring sizes?
(7) Are the analyses in paragraph 3 and following are based on the FEM or the inversion method?
(8) The three absorption peaks arise from the three different ring sizes and seems to be pretty much independent in first approximation. Can there be a coupling by polarization effects (i.e. some kind of cross talk between the rings). Are these kind of effects included in the simulation?
(9) What is the proposed mechanism to change the carrier mobility in the graphene layer? (doping) What is a realistic scenario to reach the values taken for the simulation?
(10) Paragraph 4: The effect of the graphene layer should be discussed in detail.
(11) Page 9, Line 190: What is meant by "absorptance peak mobility"?
(12) Please leave a space between the number and the unit (e.g. "0.8 µm"), since both came from different scalar spaces.
(13) References: Ref. [8] and [10] and [24]/[30] are identical, one copy can be deleted. Please check the author name in Ref. [15], since there is a number in the name
(14) Grammar:
- Page 1, Line 20: "absorption peaks" instead "absorption peak"
- Page 1, Line 22: "in high" without "to"
- Page 2, Line 51: "is" is too much in this sentence.
- Page 2, Line 55: "different sized gold rings"
- Page 2, Line 70: "[...] is negligible and therefore fixed at epsilon_SiO2=3.9 [...]"
- Page 3, Line 79: Space is missing between "h bar" and "and"
- Page 7, Line 148: "Absorptance"
- Page 7, Line 152: "red-shifted by"
- Page 7, Line 153: "decreased"
- Page 8, Line 163: "[...] first two absorptance peaks [...]"
Round 2
Reviewer 1 Report
The authors have presented all the answers accordingly. I recommend this manuscipt for possible publication.
Reviewer 2 Report
I would like to thank the authors for answering all my questions and editing the manuscript.
Some minor language mistakes should be corrected before publication:
- Line 35: "Nonetheless"
- Line 51: "[...] seems promising."
- Line 60: "[...] can be tuned [...]"
- Line 119: "electromagnetic"
- Line 154: "graphene"
- Line 180: "[...] proposed by us includes a layer [...]"